# A Cellular Assay for the Identification and Characterization of Connexin Gap Junction Modulators

**DOI:** 10.3390/ijms22031417

**Published:** 2021-01-31

**Authors:** Azeem Danish, Robin Gedschold, Sonja Hinz, Anke C. Schiedel, Dominik Thimm, Peter Bedner, Christian Steinhäuser, Christa E. Müller

**Affiliations:** 1PharmaCenter Bonn, Pharmaceutical Institute, Pharmaceutical & Medicinal Chemistry, Rheinische Friedrich-Wilhelms-Universität Bonn, An der Immenburg 4, D-53121 Bonn, Germany; adanish32@googlemail.com (A.D.); rgedscho@uni-bonn.de (R.G.); shinz@uni-bonn.de (S.H.); schiedel@uni-bonn.de (A.C.S.); dthimm@uni-bonn.de (D.T.); 2Institute of Cellular Neuroscience, Medical Faculty, Rheinische Friedrich-Wilhelms-Universität Bonn, Venusberg-Campus 1, D-53127 Bonn, Germany; Peter.Bedner@ukbonn.de

**Keywords:** compound library, connexin-43, gap junctions, GloSensor luciferase, HeLa cells, screening

## Abstract

Connexin gap junctions (Cx GJs) enable the passage of small molecules and ions between cells and are therefore important for cell-to-cell communication. Their dysfunction is associated with diseases, and small molecules acting as modulators of GJs may therefore be useful as therapeutic drugs. To identify GJ modulators, suitable assays are needed that allow compound screening. In the present study, we established a novel assay utilizing HeLa cells recombinantly expressing Cx43. Donor cells additionally expressing the Gs protein-coupled adenosine A_2A_ receptor, and biosensor cells expressing a cAMP-sensitive GloSensor luciferase were established. Adenosine A_2A_ receptor activation in the donor cells using a selective agonist results in intracellular cAMP production. The negatively charged cAMP migrates via the Cx43 gap junctions to the biosensor cells and can there be measured by the cAMP-dependent luminescence signal. Cx43 GJ modulators can be expected to impact the transfer of cAMP from the donor to the biosensor cells, since cAMP transit is only possible via GJs. The new assay was validated by testing the standard GJ inhibitor carbenoxolon, which showed a concentration-dependent inhibition of the signal and an IC_50_ value that was consistent with previously reported values. The assay was demonstrated to be suitable for high-throughput screening.

## 1. Introduction

Gap junction (GJ) channels are important for cell-to-cell communication in most tissues. They enable the free diffusion of molecules up to a molecular weight of about 1000 Da including second messengers, amino acids, ions, glucose and other metabolites [1]. Gap junctional intercellular communication regulates embryonic development and coordinates many processes including smooth and cardiac muscle contraction, tissue homeostasis, apoptosis, metabolic transport, cell growth and cell differentiation [1,2].

Connexins constitute a multigene family whose members can be divided based on their molecular weight. In humans, 21 connexin subtypes have been identified, and more than one connexin subtype can typically be found on the same cell [3,4,5]. The oligomerization of six connexins leads to the formation of a connexon that can either be homomeric or heteromeric. The docking of two connexons of adjacent cells leads to the formation of intercellular channels that may constitute either homotypic or heterotypic, or combined heterotypic/heteromeric arrangements of GJs [2,5]. Among all connexins in humans, Cx43 is the most abundantly and widely expressed connexin type, being present in many cell types and tissues [6].

Several human diseases have been linked to germline mutations of connexin family members [1,7]. For instance mutations in Cx26 result in keratitis-ichthyosis-deafness and mutations in Cx32 cause the X-linked Charcot–Marie–Tooth disease [8,9]. Mutations in Cx43 are linked to oculodentodigital dysplasia characterized by developmental abnormalities, and total disruption of this gene causes cardiac arrhythmias [10,11,12,13,14]. Changes in the expression levels of Cx43 have been reported for some neurological disorders in humans such as epilepsy, depression, and brain metastasis, implying its crucial role in the etiology or progression of these diseases [9,10,11]. Cx43 knockout mice die shortly after birth. Moreover, mouse models with a Cx43 truncated at the *C*-terminus exhibited a defect in skin barrier function with mislocalized Cx43 GJs, which also resulted in the death of the animals soon after birth [15].

Assays for testing connexin activity have previously been developed. They are based on dye transfer or quantification of a transferred molecule using luminescence detection. In the dye transfer assays, donor cells are loaded with a membrane-impermeable dye. After incubation with acceptor cells the dye can diffuse though GJs to the acceptor cells, and the extent of this effect is a measure of GJ permeability. The assays are mostly evaluated visually, but automated approaches have also been established [16,17,18]. Picoli et al. (2019) published a dye transfer assay in a high-throughput screening (HTS) format [19]. However, sophisticated imaging instrumentation is required to automatize the evaluation. Moreover, GJ inhibitors neither displayed stable nor clear concentration-dependent effects in this assay. Other established assays, which are based on luminescence, utilize a mediator molecule selectively generated in the donor cells. This compound migrates through the GJs into the acceptor cells where it activates or quenches a luminescence-emitting protein. Because of their real-time read-out, luminescence assays are suitable for HTS approaches. Lee et al. (2015) published an HTS assay, utilizing donor cells transfected with an iodide transporter and acceptor cells transfected with the yellow fluorescent protein (YFP) [20]. After iodide is taken up by the donor cells, it migrates to the biosensor cells via the GJs. The YFP fluorescence is quenched by iodide, allowing direct measurement of GJ permeability. However, the measured fluorescence could only be reduced by up to 50%, resulting in only a small assay window, which appears unsatisfactory. Another HTS assay, introduced by Haq et al. (2013), utilizes Ca^2+^ diffusion through GJs [21]. Either a G_q_ protein-coupled adrenergic receptor or a TRPV1 ion channel was used to increase intracellular Ca^2+^ concentrations in the donor cells. The acceptor cells recombinantly expressed aequorin intracellularly, which produces luminescence in the presence of Ca^2+^. However, Ca^2+^ can induce protein kinase C-mediated phosphorylation of Cxs and calmodulin resulting in an inhibition of GJs, which limits the significance of this assay [22,23].

Thus, despite the recent development of several assays for testing and identifying GJ modulators, all reported methods are far from ideal and each is associated with a number of drawbacks. Therefore, we set out to establish a novel assay to allow the screening of compound libraries as a basis for the development of GJ modulators.

## 2. Results

### 2.1. Assay Design

In the present study, we selected cAMP as a suitable analyte which cannot penetrate cell membranes passively due to its negative charge, but is able to migrate through GJs from a donor cell to a biosensor cell [24]. A genetically engineered firefly luciferase known as cAMP GloSensor-20F (GloSensor luciferase, Promega), which can detect cAMP with high sensitivity, was selected for detecting intracellular cAMP in real time (Figure 1a). Firefly luciferase catalyzes the oxidation of its substrate luciferin in the presence of Mg^2+^, ATP and O_2_ to produce oxyluciferin, AMP, CO_2_ and a luminescence signal (Figure 1b). GloSensor luciferase is a genetically engineered form of firefly luciferase, which contains the conserved cAMP binding domain B from protein kinase A regulatory subunit IIβ (see Figure 1a). Binding of intracellular cAMP to the enzyme favors the functional luciferase conformation, which then metabolizes luciferin to oxyluciferin, thereby producing a yellow-green light [25].

Our experimental strategy was to create donor cells which coexpress the human (h) G_s_ protein-coupled adenosine A_2A_ receptor (A_2A_AR) as well as Cx43. In addition, biosensor cells expressing the cAMP-sensing GloSensor luciferase along with Cx43 were engineered. A_2A_AR activation by the A_2A_-selective agonist CGS-21680 was expected to lead to adenylate cyclase activation resulting in an increase in intracellular cAMP concentrations in the donor cells. cAMP would then migrate into the biosensor cells via Cx43 GJs and could then activate the cAMP-dependent GloSensor luciferase. This was expected to result in a quantifiable luminescence signal. The principle of the designed assay is depicted in Figure 2. To initially evaluate and validate the feasibility of the new assay, preliminary experiments were performed (see below).

### 2.2. Preparation and Evaluation of Recombinant Cells

For the experiments, we aimed at utilizing a cell line that shows no or only low native Cx43 expression. HeLa cells, an immortal cervical cancer cell line, were selected because they are known to be communication-deficient due to low or lacking Cx expression [20]. First, we confirmed very low native Cx43 expression in this cell line (see Appendix A). The recombinant expression of Cx43 in HeLa cells led to significant Cx43 levels as confirmed by fluorescence microscopy using a Cx43-specific antibody (Figure 3).

The HeLa cells were transiently transfected either with A_2A_AR and GloSensor luciferase (HeLa-A_2A_AR-GSL), or only with GloSensor luciferase (HeLa-GSL). These cell constructs were subsequently tested for their luminescence signals after activation using the selective A_2A_AR agonist CGS-21680, or the nonselective AR agonist NECA (Figure 4). Forskolin, which is a direct activator of adenylate cyclase, was employed as a positive control, while the solvent DMSO (1%) served as negative control. As expected, HeLa-A_2A_AR-GSL as well as HeLa-GSL cells produced a luminescence signal upon stimulation with forskolin (10 µM), which was completely independent of AR expression. The nonselective adenosine receptor (AR) agonist NECA showed a much weaker signal in HeLa-GSL as compared to HeLa-A_2A_AR-GSL cells. The moderate signal observed for NECA is due to native expression of A_2B_ARs [26,27], which are also Gs protein-coupled like the A_2A_ARs. In contrast, the A_2A_AR-selective agonist CGS-21680, employed at a high concentration of 100 µM, only produced a cAMP-dependent luminescence signal in HeLa-A_2A_AR-GSL, but not in HeLa-GSL cells.

Based on these findings, we concluded that native HeLa cells do not express functional A_2A_ARs. This makes the A_2A_AR an excellent cAMP-generating tool by activating it with the A_2A_AR-selective agonist CGS-21680 after recombinant expression in HeLa cells. Our results were in agreement with the previously reported expression profile of A_2A_A- and A_2B_ARs in native HeLa cells, where A_2B_AR expression was found to be higher than that of A_2A_AR, and in such cases, A_2A_ signaling is not observed since it is blocked by A_2B_ARs [26,27].

As a next step, biosensor cells were created which coexpress GloSensor luciferase and Cx43. The heterologous expression of two or more proteins in the same cell may result in reduced expression levels of each protein due to high occupancy of the cellular translational machinery. Moreover, part of the cAMP produced in the donor cells might be released into the extracellular space through Cx43 hemichannels [28]. Therefore, experiments were performed to compare transiently transfected HeLa cells expressing only GloSensor luciferase with cells coexpressing GloSensor luciferase and Cx43 with regard to the luminescence signal induced by cAMP (Figure 5). Forskolin (10 µM)-mediated activation of adenylate cyclase in HeLa cells only transfected with GloSensor luciferase resulted in similar luminescence signals as compared to HeLa cells transfected with both GloSensor luciferase and Cx43 while the A_2A_AR-selective agonist CGS-21680 (1 µM) was inactive as expected since the cells did not express functional A_2A_ARs. These results clearly showed that coexpression of Cx43 and GloSensor luciferase in HeLa cells yielded sufficiently high cAMP-dependent luminescence signals for the planned assay.

### 2.3. Assay Optimization 

To investigate GJ-mediated coupling between donor and biosensor cells, we cocultured the cells at a ratio of 3:1 (donor: biosensor cells) and incubated them at 37 °C for 4 h. This ratio of donor to biosensor cells was found to be optimal based on preliminary studies using different cell numbers and ratios. The culture medium was then replaced by an assay buffer containing luciferin (the substrate of the GloSensor luciferase) in which the cells were incubated for 2 h at room temperature. Upon activation of the A_2A_ARs, recombinantly expressed only in the donor cells by the selective agonist CGS-21680, a luminescence signal was detected. However, the assay window was very small, only slightly above the DMSO control and much smaller than that observed with forskolin (Figure 6a).

cAMP produced in the cytosol is quickly hydrolyzed by phosphodiesterases (PDEs) yielding AMP. The addition of PDE inhibitors, such as 3-isobutyl-1-methylxanthine (IBMX), allows intracellular accumulation of cAMP. We therefore wondered whether the addition of an PDE inhibitor would lead to an increased luminescence signal. Thus, cocultures of donor and biosensor cells were additionally incubated with IBMX (200 µM) for 45 min at room temperature in assay buffer, after the 4 h incubation period in cell culture medium. The subsequent activation of A_2A_ARs by CGS-21680 produced a significantly increased luminescence signal (Figure 6b).

The luminescence signal produced by the biosensor cells in response to activation of A_2A_ARs expressed in the donor cells amounted to about 40% of the signal obtained in response to forskolin.

In order to verify that the activation of the GloSensor luciferase in the biosensor cells was entirely mediated by cAMP migrating from the donor cells via Cx43 GJs, both donor and biosensor cells that were not transfected with Cx43 were studied. In this crucial experiment, donor cells (HeLa-A_2A_AR) and biosensor cells (HeLa-GSL) lacking Cx43 were cocultured in a ratio of 3:1. The same experimental procedure was performed as for the actual Cx43 GJs assay (see above). Stimulation of the cocultured, Cx43-lacking donor and biosensor cells with CGS-21680 (1 µM) did not result in any increase in the luminescence signal (Figure 7) indicating that the cAMP produced in the donor cells did not reach the biosensor cells due to the lack of Cx43 channels. Only when the cocultured cells were treated with forskolin (10 µM), a high luminescence signal was observed showing that the biosensor cells were functional. These results confirmed that native HeLa cells are devoid of efficient cell-to-cell communication, and that the cAMP generated in the donor cells could only be transported via heterologously transfected GJ proteins to the biosensor cells to yield a luminescence signal. This is in line with our findings from Western blot experiments performed to analyze endogenous Cx43 expression in HeLa cells (Appendix A). Although very low levels of endogenous Cx43 could be detected in the blots, they are apparently unable to form functional GJs in HeLa cells at levels detectable by our developed assay. 

### 2.4. Assay Validation I: Effect of the Gap Junction Blocker, Carbenoxolone

As a next step, the effect of the most commonly used Cx43 GJ blocker, carbenoxolone [29], a steroidlike derivative of the natural product glycyrrhetinic acid, was investigated. Carbenoxolone showed a concentration-dependent inhibition of the luminescence signal in the newly developed Cx GJ assay with an IC_50_ value of 44.5 ± 4.8 µM (Figure 8). This corresponds well with the literature IC_50_ values ranging from 17 to 210 µM [21,30]. No relevant cytotoxicity in the HeLa cells was observed at the employed concentrations as confirmed by the 3-(4,5-dimethylthiazol-2-yl)-2,5-diphenyltetrazoliumbromid (MTT) assay as a measure for cell viability (Appendix A).

### 2.5. Assay Validation II: Suitability for High-Throughput Screening

Next, we evaluated the suitability of the optimized assay for high-throughput screening (HTS) by calculating the screening window coefficient known as *Z*′-factor. This dimensionless factor provides valuable information on the assay window, i.e., the separation between positive and negative controls. To this end, the luminescence pre-readout of the cocultures was compared with the luminescence signal obtained upon reaching a plateau after 19 min of stimulation with CGS-21680 (1 µM). Using the following Equation (1):(1)$$Z^{\prime }=1-\frac{\left(3\sigma_{c^{+}}+\;3\sigma_{c^{-}}\right)}{\left|({µ}_{c^{+}}-{µ}_{c^{-}})\right|}$$

(*σ*, standard deviation; *µ*, mean; *c*^+^, positive control; *c^−^*, negative control), the *Z*´-factor of the current assay was calculated to be 0.5. This corresponds to an HTS assay of sufficiently good quality [31]. The assay window was high with a signal to control ratio of about 3 (Figure 9). Thus, the newly developed functional Cx GJ assay can serve as an HTS platform to identify and characterize GJ modulators.

### 2.6. Screening of a Compound Library

Finally, we employed the developed assay to screen a small compound library containing 143 bioactive molecules (part of the Tocris compound library, https://www.tocris.com/products/tocriscreen-mini_2890). The compounds were tested at a concentration of 10 µM, and hits were defined as compounds showing a 25% decrease or increase in the signal compared to control (Appendix A). Initially, eleven hits were obtained, of which ten had to be subsequently discarded. The reasons were high cytotoxicity or cAMP-inducing activity on HeLa-GSL cells. These effects resulted in false positive assay results. One Cx43 GJ-inhibiting hit remained, U-54494A hydrochloride, an experimental tool compound that is, however, no longer commercially available and could therefore not be further characterized up to now. Interestingly, the compound was described as possessing anticonvulsant properties [32].

## 3. Discussion

Connexins form GJs that allow the exchange of molecules between adjacent cells, thereby facilitating even long-distance cell communication. The functional modulation of GJs, inhibition as well as enhancement, has been proposed as a novel strategy for the treatment of various diseases. These include cardiac diseases, such as arrhythmias, remodeling after cardiac infarction, and atherosclerosis [33,34], brain diseases, e.g., epilepsy and neurodegenerative diseases [35,36], and even cancer [37]. Notably, a recent study suggested that loss of GJ coupling represents a cause of human temporal lobe epilepsy [38].To identify novel GJ modulators, a suitable assay for compound screening was required. However, the previously reported assays all appeared to be fraught with various drawbacks, such as low sensitivity, low signal-to-noise ratio, propensity to interfere with test compounds, or lacking suitability for HTS [16,17,18,19,20,21]. Therefore, we designed a completely new approach based on the sensitive measurement of luminescence by an engineered luciferase that is sensitive to the polar second messenger cAMP [25]. We selected communication-deficient HeLa cells, a permanent human cancer cell line that lacks the expression of functional GJs [39]. cAMP is produced intracellularly from ATP by the enzyme adenylate cyclase (AC). G protein-coupled receptors that activate Gs proteins lead to the stimulation of AC and thus to cAMP production [40]. This is a fast response occurring within a few seconds after receptor activation. We had previously observed that HeLa cells do not express functional A_2A_ARs, only A_2B_ARs [27]. Therefore, we recombinantly expressed A_2A_ARs along with Cx43 in HeLa cells to obtain donor cells, while biosensor cells were transfected with Cx43 and the cAMP-sensitive luciferase (GloSensor luciferase, GSL). In fact, a number of control experiments demonstrated that the cells behaved as expected. While HeLa-GSL cells did not respond to application of the A_2A_-selective AR agonist CGS-21680, the same cell line cotransfected with the A_2A_AR showed a large luminescence signal upon treatment with the A_2A_AR agonist. As expected, forskolin, a direct activator of AC that was used as a positive control, always led to a luminescence signals. Since cAMP is rapidly degraded by PDEs, we had to add a nonspecific PDE inhibitor (IBMX) prior to the experiments. A mixture of donor cells expressing A_2A_ARs plus Cx43, and biosensor cells expressing the cAMP-sensitive luciferase (GSL) plus Cx43 at a proportion of 3:1 resulted in a satisfactory signal of the A_2A_AR agonist CGS-21680 (1 µM) in the presence of IBMX. The same cells, but lacking Cx43, gave no signal under the same conditions proving that the assay worked as intended. It should be noted that an increase in cAMP concentration might itself exert modulatory effects on the Cx GJs, such as changes in permeability or increased expression of Cx. However, at least the latter effect would take much longer than the duration of the assay. The assay could finally be validated by the known GJ inhibitor carbenoxolone, which is the most frequently used GJ inhibitor that has been explored and utilized in many studies [19,21,29,30,38]. We determined an IC_50_ value of 44.5 µM, which is good in agreement with the literature data [21,30] and which was demonstrated to be not due to cell toxicity as evidenced by its results in an MTT assay performed to assess cell viability (see Appendix A). Our new assay was also found to be very suitable for HTS, with a *Z*′ value of 0.5 [31]. Finally, a first compound library was screened, leading to the discovery of the first hit compounds. In future studies, resynthesis and characterization of the most promising hit compound will have to be performed. The assay is also ready for the screening of further compound libraries to identify novel modulators of Cx43 GJs that can subsequently be optimized by medicinal chemistry approaches. The newly developed assay is not limited to Cx43, but can be expected to have the potential for broad application by adapting it to other members of the Cx family.

## 4. Materials and Methods 

### 4.1. Cultivation of Cells

HeLa cells with a low endogenic connexin expression were a gift of K. Willecke. They were cultivated in an appropriate cell culture flask with culture medium consisting of Dulbecco’s Modified Eagle Medium (DMEM, Thermo Fisher Scientific, Waltham, MA, USA) supplemented with 10% fetal calf serum (Sigma-Aldrich, Darmstadt, Germany), 100 U/mL penicillin G, and 100 µg/mL streptomycin (Thermo Fisher Scientific). Medium for transfected cells was additionally supplemented with 800 µg/mL G418 (Merck KGaA, Darmstadt, Germany). All steps were performed under sterile conditions (laminar air flow hood). The cultures were incubated at 37 °C in a humidified atmosphere with 5% CO_2_ in an incubator. Cells were regularly passaged after they had reached a confluence of 80–90%. The old medium was taken off and the cells were washed twice with sterile PBS to remove residual medium. The cells were detached using a trypsin 0.01% (Lonza Group Ltd., Basel, Switzerland) solution containing 0.6 mM EDTA (Carl Roth GmbH + Co. KG, Karlsruhe, Germany) with incubation at 37 °C for 2–3 min. 

### 4.2. Buffer Preparation for Cell Culture (PBS)

NaCl (137 mM), KCl (2.5 mM), Na_2_HPO_4_ (7.5 mM), and KH_2_PO_4_ (1.5 mM) were dissolved in deionized water and pH was adjusted to 7.4 with HCl (37%). The buffer was autoclaved and stored at room temperature.

### 4.3. Expression Vectors and Molecular Cloning

The coding region of the human A_2A_AR was subcloned into the retroviral vector pQCXIN into the NotI and BamHI restriction sites. The following primers were designed: 

f-hA_2A_-NotI: 5′-gtgacagcggccgcatgcccatcatgggctcctc-3′ and r-hA_2A_-BamHI: 5′- cttactaggatcctcaggacactcctgctccatc-3′. The following PCR program using Pyrobest™ DNA polymerase (Takara Bio, Mountain View, CA, USA) was applied: 10 s at 98 °C and 30 cycles consisting of 10 s at 98 °C, 30 s at 62–66 °C, and 1 min at 72 °C followed by a final elongation step of 10 min at 72 °C. The PCR product was purified and digested with NotI and BamHI. The retroviral vector pQCXIN was cut with NotI and BamHI and after purification it was ligated with the cut PCR product. The correct assembly of the gene was verified by sequencing (Eurofins, Ebersberg, Germany). 

The plasmid encoding for the modified firefly luciferase pGloSensor™-20F cAMP was purchased from Promega (Madison, WI, USA). 

The plasmid encoding for the mouse Cx43 was kindly provided by Prof. Dr. Klaus Willecke, Life and Medical Sciences Institute (LIMES), University of Bonn. The coding region of the mouse Cx43 was subcloned with XhoI and XbaI into the pcDNA™4/myc-His A vector, which was cut with XhoI and XbaI. The following primers were designed: 

f-mCx43-XhoI: 5′- GAGCTACTCGAGACCATGGGTGACTGGAGCGCC-3′, 

r-mCx43-XbaI: 5′- CATCATTCTAGATTAAATCTCCAGGTCATCAGGCCGAGG-3′. 

The PCR was conducted as described above. The PCR product was purified and digested with XhoI and XbaI and ligated with the prepared vector pcDNA™4/myc-His A. The correct assembly of the gene was verified by sequencing (Eurofins, Ebersberg, Germany).

### 4.4. Retroviral Transfection

Packaging cells GP+envAM12 (1.5 × 10^6^ cells) (LGC Standards GmbH, Wesel, Germany) were seeded in a 25 cm^2^ flask with 5 mL of culture medium followed by incubation (37 °C, 5% CO_2_, 24 h) prior to transfection. Next day the cells were transfected using Lipofectamine 2000 (Thermo Fisher Scientific) with 10 µg of DNA comprising the retroviral plasmid (6.25 µg) and VSV-G plasmid (3.75 µg). After 15 h of transfection, the old culture medium was exchanged with 3 mL of fresh culture medium supplemented with 5 mM sodium butyrate (Sigma-Aldrich) followed by incubation (32 °C, 5% CO_2_, 48 h). The supernatant (3 mL) containing the virus particles was removed and sterile filtered using a 2 µm filter to harvest the viruses. The filtrate containing the viruses was mixed with 6 µl of polybrene solution (4 mg/mL in H_2_O, sterile filtered). Subsequently, the medium of the target cell line (HeLa) within a 25 cm^2^ flask was replaced with the mixture and the cells were incubated (37 °C, 5% CO_2_, 2.5 h). After the incubation, the mixture containing the viruses was discarded and 5 mL of fresh culture medium supplemented with G418 was added to the cells followed by incubation (37 °C, 5% CO_2_, 48–72 h). After three days of incubation, the medium was changed until the nontransfected cell death process ended.

### 4.5. Lipofectamine Transfection (Lipofection)

For transfection, cells (2 × 10^6^) were seeded in a 25 cm^2^ culture flask containing culture medium followed by incubation (37 °C, 5% CO_2_, 16 h). The old culture medium was exchanged against 6.25 mL of new full culture medium without antibiotics and the culture was incubated (37 °C, 5% CO_2_, 3 h). Basal medium (600 µL) without any supplements was mixed with 25 µL of Lipofectamine 2000 (Thermo Fisher Scientific) and incubated for 5 min at RT. Plasmid-DNA (10 µg) was diluted in basal medium without any supplements to make a final volume of 625 µL. Both solutions were mixed giving a mixture of 1225 µL, which was incubated for 20 min at room temperature. The transfection mixture was then dropwise added to the cells followed by incubation (37 °C, 5% CO_2_, 24–48 h).

### 4.6. Sample Preparation for Fluorescence Microscopy

Cells were seeded in 6 well plates. Prior to the cells, one sterile microscopy coverslip was added to each well. After one day of incubation (37 °C, 5% CO_2_) the coverslips were washed in PBS and incubated with 4% paraformaldehyde/PBS for 20 min at room temperature. Subsequently cells were incubated with PBS + 1% BSA for 15 min, with the primary antibody for 60 min, with the secondary antibody for 30 min and with DAPI for 5 min. Between every step, the coverslips were washed with PBS + 1% BSA. Incubation steps were performed at room temperature in the dark. One drop of mounting medium (Fluoromount) was placed on a slide and the coverslip containing the cells was placed onto the drop. The sample was left to dry overnight in the dark at 4 °C.

### 4.7. Characterization of Recombinant Cells

HeLa cells were transiently transfected with the A_2A_AR and GloSensor luciferase (HeLa-A_2A_AR-GSL) or only with GlosSensor luciferase (HeLa-GSL). The cells were seeded into wells (60,000/well) and incubated overnight. The growth medium was replaced with medium supplemented with 2% GloSensor luciferase reagent and incubated for 2 h at 37 °C. After addition of the compounds dissolved in DMSO the cells were incubated at 37 °C for 15 min. The luminescence was subsequently measured in a plate reader (Mithras LB 940, Berthold Technologies, Bad Wildbad, Germany) without any filters.

### 4.8. Buffer Preparation for Cx43 GJ Assay (HBSS + BSA)

HEPES (20 mM), NaCl (137 mM), glucose (5.5 mM), KCl (5.4 mM), NaHCO_3_ (4.2 mM), CaCl_2_ (1.25 mM), MgCl_2_ (1 mM), MgSO_4_ (0.8 mM), KH_2_PO_4_ (0.44 mM) and Na_2_HPO_4_ (0.34 mM) were dissolved in autoclaved water and pH was adjusted to 7.4. The buffer was stored at 4 °C. BSA (0.1%, *w*/*v*) (AppliChem, Darmstadt, Germany) was dissolved in buffer prior to use.

### 4.9. Optimized Cx43 Gap Junction Assay

Donor cells were retrovirally transfected with A_2A_ARs and biosensor cells with GloSensor-20F (Promega, Madison, WI, USA), respectively. Both cell lines were transfected with Cx43 via lipofection.

On the day of the experiment, donor and biosensor cells were harvested by trypsinization, centrifuged, and resuspended in culture medium. Cell aggregates were dislodged by slow pipetting and cells were counted. Initially, 30,000 biosensor cells per well were dispensed in a 96-well solid bottom white plate followed by 90,000 donor cells to maintain a 3:1 ratio of donor and biosensor cells. Both donor and biosensor cells were mixed by pipetting and the coculture was incubated for 4 h at 37 °C, 5% CO_2_. After the incubation, the full DMEM medium was replaced by HBSS buffer supplemented with 0.1% BSA, 200 µM IBMX and 2% GloSensor cAMP reagent (Promega, Madison, WI, USA) (“assay buffer”) followed by incubation for 1 h in the dark at room temperature. For the evaluation of GJ inhibitors, the compounds were added directly when DMEM full medium was replaced with the assay buffer. The plate was then placed in a plate reader (Mithras LB 940, Berthold Technologies, Bad Wildbad, Germany) without any filter for the basal luminescence readout. Subsequently, either CGS-21680 (10 µM) (Bio-Techne GmbH, Wiesbaden-Nordstadt, Germany), DMSO (1%) or forskolin (10 µM) (AppliChem, Darmstadt, Germany) was added, and luminescence from each well was measured, either in a kinetic mode for 1 s with an interval of 3 min over a total duration of 30 min, or as a single point after 15–20 min.

### 4.10. Data Evaluation and Statistical Analyis

The data were analyzed using Prism 8.0 (GraphPad Software Inc., San Diego, CA, USA). Differences between means were tested for significance by 2-way ANOVA or repeated measures 2-way ANOVA and Dunnet’s multiple comparisons test. The NIS Element Advanced Research software 4.0 was used for microscopy image acquisition and analysis.

## Figures and Tables

**Figure 1 ijms-22-01417-f001:**
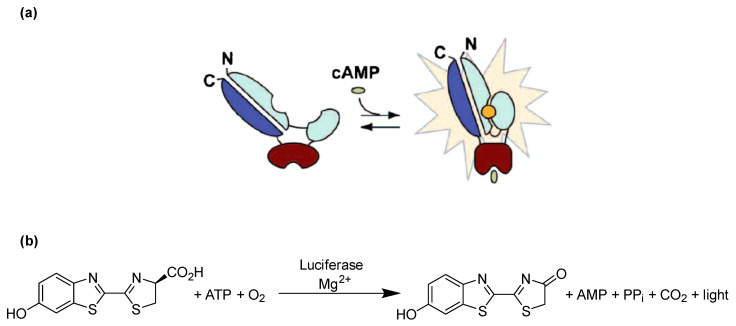
Principle of cAMP detection using the engineered GloSensor luciferase. (**a**) GloSensor luciferase in the open conformation shows negligible activity resulting in a low luminescence background, whereas binding of cAMP to the cAMP binding site favors the closed conformation and hence activates the luciferase, which metabolizes luciferin to oxyluciferin, producing luminescence. (**b**) Luciferase catalyzes the oxidation of luciferin using molecular oxygen and ATP in the presence of Mg^2+^ to produce oxyluciferin, which is highly unstable in an electronically excited state and produces light upon returning to its electronical ground state. Modified based on published figure [25].

**Figure 2 ijms-22-01417-f002:**
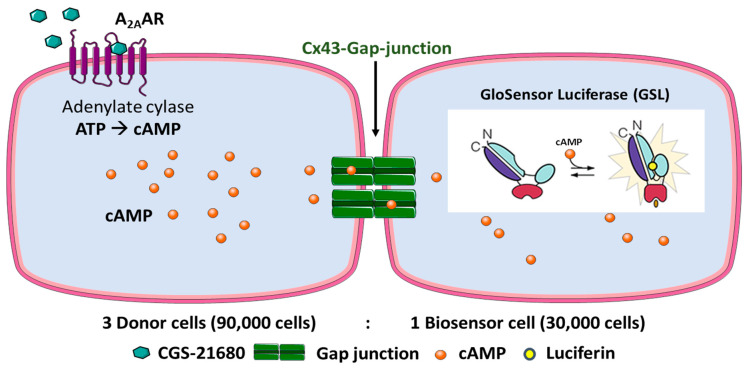
Design of the Cx43 GJ assay. HeLa cells expressing A_2A_AR and Cx43 are denoted as donor cells and HeLa cells expressing GloSensor luciferase and Cx43 as biosensor cells. The cell lines were cocultured in a ratio of 3:1 (donor: biosensor cells) for 4 h to allow the formation of Cx43 GJs. The cells were then equilibrated with buffer containing the substrate (luciferin) of the engineered cAMP-dependent luciferase. Upon activation of the G_s_ protein-coupled A_2A_ARs, cAMP levels in the donor cells are increased. cAMP can then migrate via the Cx43 GJs to the biosensor cells. There, cAMP binds to the GloSensor luciferase which results in a conformational change that leads to an activation of the GloSensor luciferase, creating a luminescence signal by oxidation of luciferin. Depiction of GloSensor luciferase is based on reference [25].

**Figure 3 ijms-22-01417-f003:**
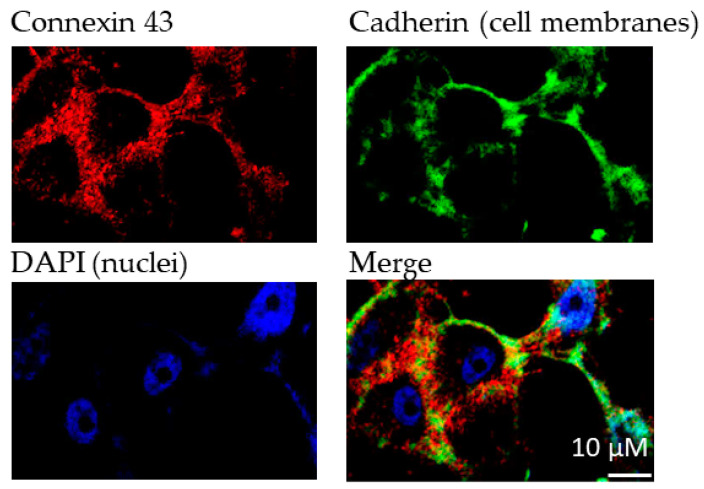
Immunofluorescence analyses of Cx43 expression in transfected HeLa cells. Cadherin was used to stain cell membranes. Primary antibodies (1:1000): antipan cadherin (mouse), anti Cx43 (rabbit); secondary antibodies (1:500): Alexa 488 (antimouse), Alexa 594 (antirabbit). DAPI (1:10,000) was used to stain cell nuclei.

**Figure 4 ijms-22-01417-f004:**
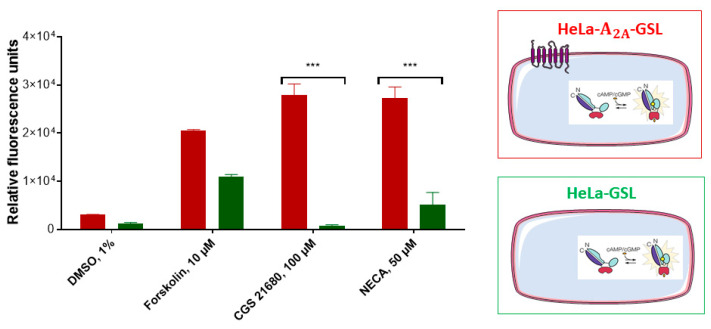
Assessment of adenosine receptor-mediated cAMP production in A_2A_AR-transfected (red) and nontransfected (green) HeLa- cells. The cAMP-activated GloSensor luciferase was cotransfected resulting in cAMP-dependent luminescence signals. Cells (60,000/well) were incubated with medium supplemented with 2% GloSensor luciferase reagent for 2 h at 37 °C. After addition of DMSO or compound dissolved in DMSO (forskolin, 10 µM, CGS-21680, 100 µM, or NECA, 50 µM) the cells were incubated at 37 °C for 15 min. Signals induced by adenosine receptor agonists CGS-21680 or NECA were significantly different between both cell lines (*** *p* < 0.001, 2-way ANOVA). For details see Section 4.

**Figure 5 ijms-22-01417-f005:**
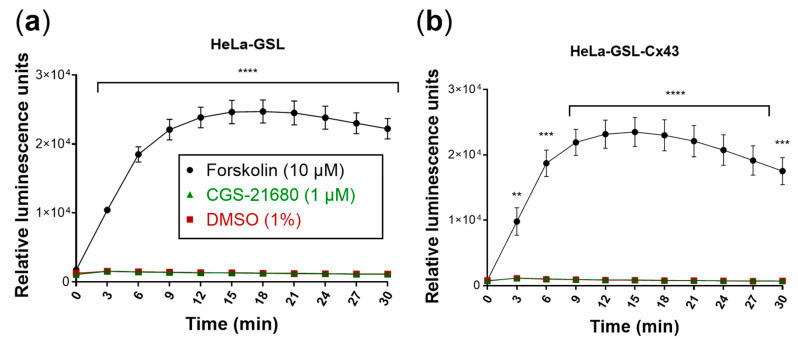
Evaluation of the biosensor cells. Forskolin (10 µM) was used as a positive control and DMSO (1%) as a negative control. Means ± SEM of three individual experiments performed in duplicates are given. (**a**) Biosensor cells produced luminescence only in response to forskolin (10 µM). Luminescence response to CGS-21680 (1 µM) was not different from control (DMSO). (**b**) Cx43 transfection of biosensor (HeLa-GSL) cells did not affect luminescence responses. Statistical significance calculated with repeated measures 2-way ANOVA and Dunnet’s multiple comparisons test comparing treatments to control (DMSO, 1%). ** *p* < 0.01, *** *p* < 0.001, **** *p* < 0.0001.

**Figure 6 ijms-22-01417-f006:**
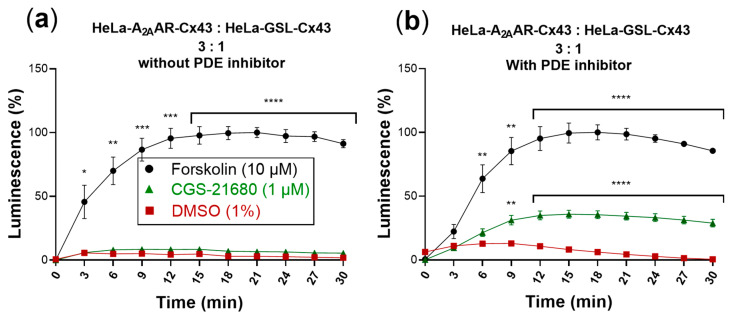
cAMP-dependent luminescence signal due to activation of cocultures of donor and biosensor cells (3:1) by the selective A_2A_AR agonist CGS-21680 (1 µM). Forskolin (10 µM) was used as positive control and DMSO (1%) served as negative control. Data are normalized to the maximal effect of forskolin (100%). Data represent means ± SEM of three individual experiments performed in duplicates. (**a**) Activation without PDE inhibitor. Only forskolin displayed a significantly increased luminescence signal compared to control. (**b**) Activation in the presence of the PDE inhibitor IBMX (200 µM). Statistical significance calculated with repeated measures 2-way ANOVA and Dunnet’s multiple comparisons test comparing treatments to control (DMSO, 1%). * *p* < 0.05, ** *p* < 0.01, *** *p* < 0.001, **** *p* < 0.0001.

**Figure 7 ijms-22-01417-f007:**
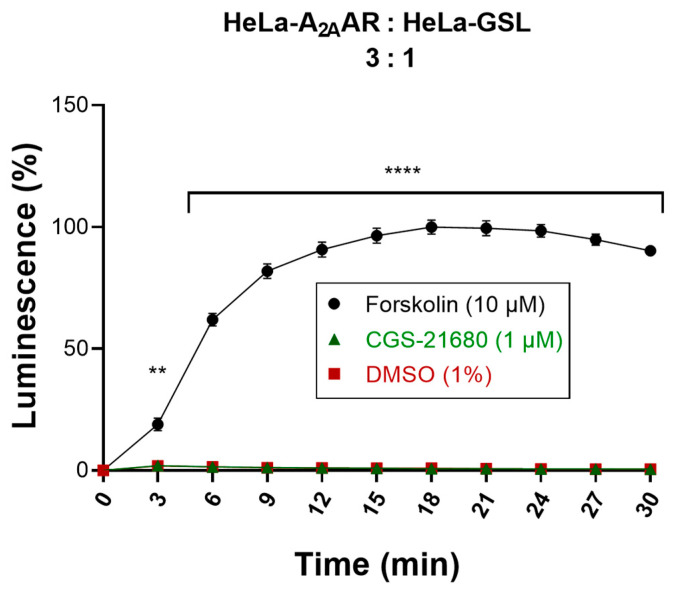
Evaluation of cocultured donor and biosensor cells lacking Cx43 GJs. Activation of cocultures of Cx43-deficient donor and biosensor cells by the selective A_2A_AR agonist CGS-21680. Forskolin (10 µM) was used as a positive control and DMSO (1%) served as a negative control. Data are normalized to the maximal effect of forskolin (100%). Data represent means ± SEM of three individual experiments performed in duplicates. Statistical significance calculated with repeated measures 2-way ANOVA and Dunnet’s multiple comparisons test comparing treatments to control (DMSO, 1%). ** *p* < 0.01, **** *p* < 0.0001.

**Figure 8 ijms-22-01417-f008:**
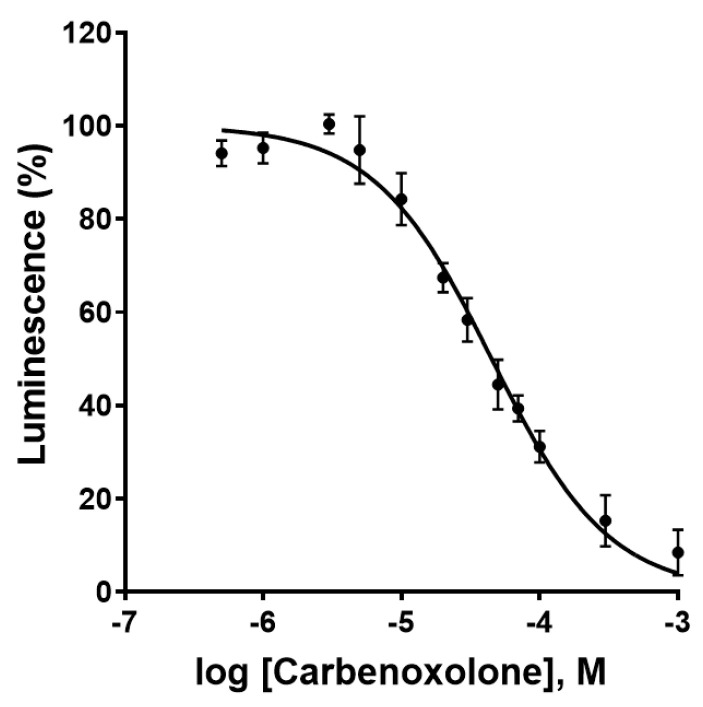
Concentration-dependent inhibition of Cx43 gap junctions by the blocker carbenoxolone as determined in the newly developed assay. Data points represent means ± SEM from 3 separate experiments. IC_50_ = 44.5 ± 4.8 µM.

**Figure 9 ijms-22-01417-f009:**
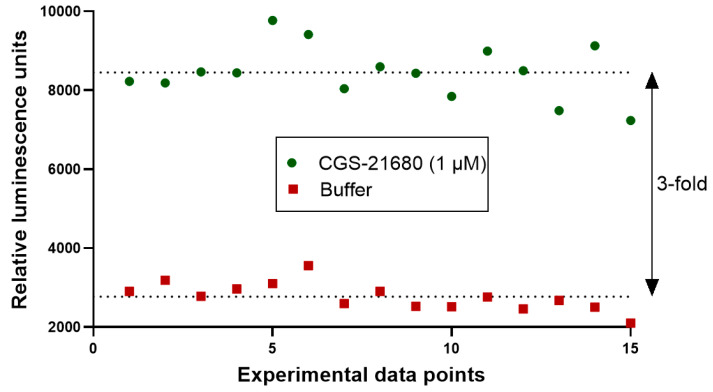
Evaluation of suitability of the newly developed Cx43 GJ assay for high-throughput screening (HTS). The quality of the assay was assessed by calculating the screening window coefficient (Z´-Factor) as previously described [31]. A coculture of donor cells (90,000 cells/well) and biosensor cells (30,000 cells/well) at a ratio of 3:1 was incubated in assay buffer for 45 min. Prior to the addition of the A_2A_AR agonist CGS-21680 (1 µM) to the coculture, basal luminescence was measured (red data points, negative control). The coculture was then activated by the addition of CGS-21680 (1 µM). The luminescence signal had reached a stable plateau 19 min after stimulation, and data points were measured as positive controls (green data points). We measured 15 separate data points for positive and negative controls, indicating reproducibility and an assay window of about 3-fold.

## Data Availability

The data presented in this study are available in the article or in Appendix A.

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
