# Peer review of "A Cellular Assay for the Identification and Characterization of Connexin Gap Junction Modulators"

_ijms, 2021, doi:10.3390/ijms22031417_

Round 1
Reviewer 1 Report
The manuscript by Mueller and coworkers, describes a novel assay for gap junction mediated intercellular communication. The intercellular flux of cAMP mediated by gap junctions is being measured by co-culturing two populations of cells, one expressing the cAMP generating adenosine receptor as donor and the other expressing a cAMP sensitive luciferase construct as recipient. Both cell types in addition express the gap junction protein connexin43. The assay is elegant and promises to be a valuable tool to screen libraries for potential modulators of gap junction function. This is important for two reasons: 1. Presently, the repertoire of gap junction drugs is very limited and most of them exhibit low specificity. 2. A large number of loss or gain of function mutations in gap junction proteins are known to cause disease. Although a variety of assays of gap junction function are available, the reliable ones are not suitable for high throughput screening. Thus, the assay described here promises to advance the field considerably.
Specific critique (minor):
- The authors have the tendency to cite recent reviews or secondary papers instead of the original findings. For example, cAMP permeability of connexin based channels needs better documentation.
- The authors correctly point out that calcium-based assays can be erroneous, because Ca2+ can induce connexin phosphorylation via protein kinaseC or Ca/calmodulin mediated inhibition of gap junctions. In fairness, the authors should discuss the possibility that the new assay also could be biased by cAMP-mediated modulation of connexin function and expression.
- Figure 4: Why is Forskolin less effective in HeLa-GSL cells?
- Page 11, line359: Is this solution calcium-free? Why? Does this not risk the opening of gap junction hemichannels?
Reviewer 2 Report
Comments for Authors
This paper entitled “A Cellular Assay for the Identification and Characterization of Connexin Gap Junction Modulators” deals with the establishment of a suitable assay to identify GJ modulators useful as potential therapeutic drugs. They used HeLa cells recombinantly expressing Cx43, creating donor cells (also expressing the Gs protein-coupled adenosine A2A receptor), and biosensor cells (expressing a cAMP-sensitive GloSensor luciferase). Once intracellular cAMP production is triggered in the donor cell, cAMP can migrate via the Cx43 gap junctions to the biosensor cells and the cAMP-dependent luminescence signal can be measured. The Authors optimized and validated their model and demonstrated its suitability for high-throughput screening.
Although I don't feel qualified to judge about the English language and style, I think this paper is well written and correct. The presentation of data is clear and convincing, and the results shown in the figures are fine
However, I have to raise some minor points that should be addressed to improve the quality of this paper.
-Abstract: I would reinforce also at the end of the abstract (line 26-27), as a concluding remark, that “The assay was demonstrated to be suitable for high-throughput” and thus useful to identify and characterize GJ modulators.
-Legend fig 5: please specify the meaning of *
-line 191:-how was the ratio 3:1 of the cells used in the co-culture chosen?
-line 212: *** not necessary, please delete it.
-Why not to show fig s3 in the main text? (I was actually not able to enter the supplementary materials section). Since the title claims “ A Cellular Assay for the Identification and Characterization of Connexin Gap Junction Modulators” I think that showing the figure of the identification of possible GJ modulators in the results section (2.6) would adequately reinforce the aim of the research.
-The first three lines of the discussion sound repetitive and should be shortened.
-Although the first part of the paper is well planned, organized and presented, in my opinion the discussion section seems to recapitulate excessively the results while lacking more critical comments about the findings. The reference to the figures in this section should be avoided. The relevance of this findings should be further strengthened also in the respect to the state of the art as well as the potential development and application of this study.
-Line 435: is it really NaCl 13 mM?
-Figures 4-7 show some statistical significance, with some indications in the figure captions. However, I think that Authors should include a Statistical analysis paragraph under the Material and Methods section, to better explain how the analysis was conducted and which tests and post hoc test were used.
